# Distribution Characteristics and Ecological Risk Assessment of Nitrogen, Phosphorus, and Some Heavy Metals in the Sediments of Yueliang Lake in Western Jilin Province, Northeast China

**Jiali Zhang [1], Yinghong Liu [1,\*], Meilan Wen [1], Chaojie Zheng [2], Sheli Chai [3], Liangliang Huang [4]** and **Panfeng Liu [1,\*]**

1. College of Earth and Sciences, Guilin University of Technology, Guilin 541006, China
2. School of Resources and Environmental Engineering, Hefei University of Technology, Hefei 230009, China
3. College of Geo-Exploration Science and Technology, Jilin University, Changchun 130012, China
4. College of Environmental Science and Engineering, Guilin University of Technology, Guilin 541006, China
* Correspondence: 6611053@glut.edu.cn (Y.L.); panfengliu@glut.edu.cn (P.L.);
  Tel.: +86-187-7735-7006 (Y.L.); +86-187-7883-2959 (P.L.)

**Abstract:** This study seeks to clarify the content characteristics, spatial distribution, potential sources and ecological risks of nitrogen, phosphorus and some heavy metals (As, Hg, Cd, Cr, Cu, Pb, Zn and Ni) in the sediments of Yueliang Lake. Nitrogen, phosphorus and heavy metals were analyzed in the surface and core sediments of Yueliang Lake. The present situation of heavy metal pollution and the degree of potential ecological risk in sediments was evaluated by the geo-accumulation index ($I_{geo}$) and potential ecological risk index ($RI$). The correlation (CA) and principal component analysis (PCA) methods were used to analyze the potential sources of the main pollutants among the heavy metals. The results show that the total nitrogen ($TN$ = 2305 mg/kg) and total phosphorus ($TP$ = 530 mg/kg) in the surface sediments of Yueliang Lake are at medium and low levels, respectively. The average content of organic matter was 2.17%, and the nutrient ratio was 6.90–11.92, which was significantly higher in the northwest than in the middle and east of Yueliang Lake, indicating that the organic matter was a mixture of endogenous aquatic plants and exogenous terrestrial plants in the sediments. From two evaluation indices ($RI$ and $I_{geo}$) calculated using element contents, the heavy metals in the surface sediments were at a moderate ecological risk level. The level of Hg was moderately polluted, Pb and Cd were at the mildly polluted level, and Cu, Zn, As, Cr and Ni were at pollution-free levels. Except for Hg, the other elements in the core sediment are basically not polluting, and the whole is at the level of slight ecological risk. The sources of heavy metals in the sediments are roughly divided into three categories. The first category is natural sources, including Cr, Ni, As, Zn and Cu. The second category includes Cd and Hg and the main sources are highly related to energy development and agricultural activities. The third category is light Pb pollution caused by vehicular traffic and coal-related industrial activities. Therefore, the pollution problems caused by tourism development and agricultural activities should be considered in the future development of the Yueliang Lake area.

**Keywords:** heavy metals; lake sediment; ecological risk; source analysis; Yueliang Lake

## 1. Introduction

Lake wetlands play an important role in conserving water sources, providing drinking water for humans and providing and managing water resources for various ecosystems [1–5]. With the acceleration of industrialization and urbanization, industrial activities such as electroplating, metal smelting, fossil fuel combustion and chemical industry wastewater discharge will release heavy metals such as As, Cu and Pb [6–9]; and agricultural production and human activities will release a large number of nutrient elements, which enter rivers through rainfall, runoff and other channels, and finally enter lakes [10,11]. These pollutants can degrade water

quality and reconstruct biological communities through physical and chemical interactions, lead to severe lake eutrophication and reduce the survival of invertebrates and fish diversity in the aquatic ecosystem [12]. As a kind of non-degradable pollutants, the heavy metals will be mostly accumulated in the sediment after entering the lake through flocculation or sedimentation. When the environmental conditions change, the heavy metals will be released from the sediment to the water again, posing a potential threat to aquatic organisms and human health [13]. Previous studies have pointed out that nutrient elements (N and P) are the main factors limiting the growth of phytoplankton in lakes [14,15], while sediments are the main endogenous load of nutrients in lakes [16,17], and the stored N and P are easily released by the processes of decomposition and analysis, thus aggravating the degree of water nutrition [18,19]. Waajen et al. believed that there was a significant correlation between lake eutrophication and lake sediments, and the release of pollutants in the sediments is the fundamental reason for the deterioration of lake water quality [20]. Heavy metals in sediments have attracted worldwide attention due to their toxicity, persistence in the environment and bioaccumulation after entering the food chain. Among them, Cd, Cr, As, Hg, Pb, Cu, Zn and Ni heavy metals have been listed as priority pollutants by the US Environmental Protection Agency [21]. Therefore, it is of great significance to investigate the nitrogen, phosphorus and heavy metals in lake sediments and evaluate the pollution degree and potential ecological risks for improving the state and safety of water environment.

The western Jilin Province is located in the hinterland of Songliao Plain and belongs to the semi-arid temperate climate zone. The ecological environment is in the fragile ecotone of agriculture and animal husbandry. There are many lakes in the area, about 700, among which Chagan Lake and Yueliang Lake are the largest and second largest lakes, respectively. As a natural ecological barrier in western Jilin Province, Yueliang Lake performs the functions of regulating climate, preserving biodiversity, flood storage and disaster reduction, agricultural irrigation, tourism development and aquaculture. Meanwhile, Yueliang Lake is also a famous tourist attraction in the western part of Jilin Province. Therefore, the water quality of Yueliang Lake has a direct impact on the production of agriculture, animal husbandry and fishery, economic development, and the health of residents in Western Jilin. In recent years, the precipitation in the Yueliang Lake area has decreased year by year, and water conservancy projects such as Chaersen have been completed in the upper reaches of Taoer River. As a result, the incoming water into the Yueliang Lake is being reduced continuously, causing the lake to shrink and the water quality to decline [22]. Therefore, the investigation and evaluation of the sediments of Yueliang Lake is of great significance for the healthy development of economy and society and the protection of the ecological environment in the area. Previous studies have only studied water nutrient status, water quality evolution and water supply [23,24], as well as the relationship between the change of deposition rate and environmental changes and human activities in different degrees [25], while the content, distribution and ecological risk assessment of heavy metals in the sediments of Yueliang Lake are rarely studied. This study takes the sediments of Yueliang Lake as the research object, analyzes the content characteristics of nitrogen, phosphorus, and heavy metals in the sediments, evaluates the pollution intensity of heavy metal using the geo-accumulation index ($I_{geo}$) and the potential ecological risk index (RI), and distinguishes the possible sources of the heavy metals, which has important supporting significance and obvious social benefits for ecological protection and restoration in the area.

## 2. Materials and Methods

### 2.1. Study Area

Yueliang Lake is in the middle of Songnen Plain, which is a fault basin of the Mesozoic and Cenozoic eras, and is located at the junction of Daan and Zhenlai counties in the northwest of Jilin Province (E 123° 42′—124° 02′, N 45° 39′—45° 48′) (Figure 1a,b). It is the largest lake in Da'an County, 37 kilometers from the center of Da'an County. The water area of the Yueliang Lake is 206 km$^2$, with a length of 25 km from east to west, a width of 10 km from north to south, average water depth of 4 m, storage capacity of 487 million m$^3$

and a swamp area of 84 km². Yueliang Lake is located in the semi-arid temperate climate zone in the middle temperate zone, with more rain and warm climate in summer, and a dry and cold climate in winter. The average annual temperature is 4.3 °C, the average annual precipitation is 411.2 mm, the average evaporation is 1756.9 mm and the relative humidity is 62%. The main recharge water sources of Yueliang Lake are Taoer River, groundwater and atmospheric precipitation. However, due to a continuing lack of rainfall in the region and the seasonal drying-up of rivers, the wetland and water areas have been shrinking sharply in recent years. The soil types around Yueliang Lake are mainly meadow swamp soil and humus swamp soil.

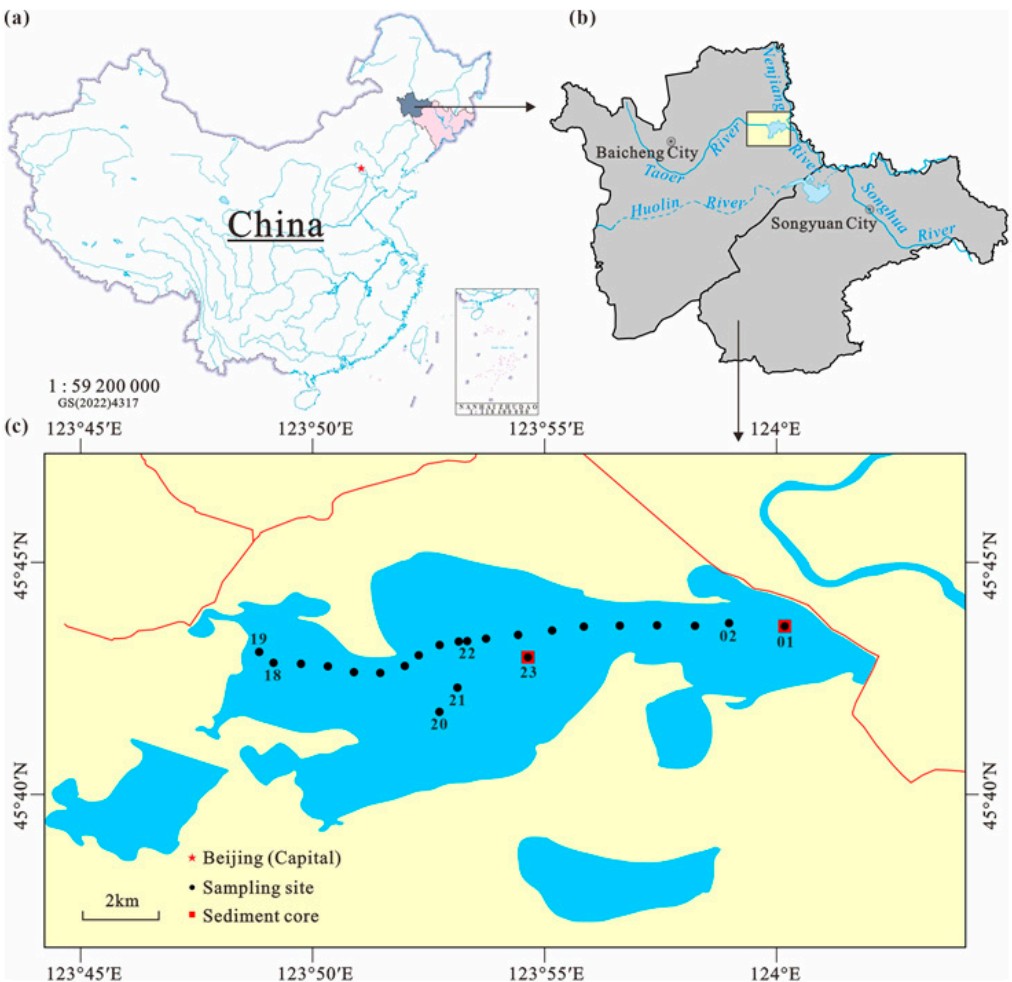

**Figure 1.** Geographic setting of Yueliang Lake (**a,b**) and detailed view of sampling sites (**c**).

### 2.2. Sample Collection and Analysis

Twenty-two surface sediment samples were collected from Yueliang Lake using a grab mud snapper, as illustrated in Figure 1c. The surface sediment samples (0~20 cm) were collected with a plastic spoon and stored in a polyethylene plastic bucket. Two sediment profiles were collected by gravity sampler at the entrance and center of Yueliang Lake (Figure 1c). The sediment core taken, sectioned at 4 cm intervals, and a total of 97 samples were stored in polyethylene bags at room temperature. All returned samples were first dried in the laboratory and then placed in the oven to dry for 24 h at 60°. After that, the dried bulk samples were ground with an agate mortar and sieved at the <120-mesh fraction for further geochemical analysis.

All samples were digested using a mixture of $HNO_3$ (15 mL)–HCl (10 mL)–HF (10 mL)–$HClO_4$ (5 mL) for the determination of the total concentrations of Ni, Cr, Cd, Pb, Cu and Zn by ICP-AES and ICP-MS, while the As and Hg concentrations were deter-

mined after aqua regia (3:1 HCl and $HNO_3$) by atomic fluorescence spectrometer (AFS). Total nitrogen and organic matter were determined by volumetric method. The analysis quality was controlled by using the national primary reference materials and duplicate samples, and the error of each element analysis was within 5%.

### 2.3. Contamination and Risk Assessment Methods

### 2.3.1. Geo-Accumulation Index

The index of geo-accumulation is a quantitative indicator which was developed in the 1970s to analyze the degree of heavy metal pollution in sediments [26], and it mainly uses the relationship between heavy metal concentration and background value to describe the degree of heavy metal pollution and environmental ecological risk [27,28]. The calculation method is shown in Equation (1):

$$I_{geo} = log_2(C_k^i / (K \times B_k^i))$$

(1)

where $I_{geo}$ represents the geo-accumulation coefficient of heavy metals; $C_k{}^i$ is the content (mg/kg) of measure element; $B_k{}^i$ is the background value of heavy metals in the area; and $K$ is the petrogenetic effect, generally set as 1.5. The $I_{geo}$ for heavy metals is classified in seven classes as the following [29]: $I_{geo} \leq 0$, uncontaminated; $0 < I_{geo} \leq 1$, slightly polluted; $1 < I_{geo} \leq 2$, moderately polluted; $2 < I_{geo} \leq 3$, moderately to heavily polluted; $3 < I_{geo} \leq 4$, heavily polluted; $4 < I_{geo} \leq 5$, severely polluted; and $I_{geo} > 5$, extremely polluted.

### 2.3.2. Potential Ecological Risk Index

The potential ecological risk index is based on the principle of sedimentology method to evaluate heavy metal pollution and ecological damage, and it can reflect the comprehensive effect of a single heavy metal or a variety of heavy metals in a particular environment, and can quantitatively reflect the degree of potential ecological harm of heavy metal pollutants [29]. The calculation method is shown in Equation (2):

$$RI = \sum E_r^i = \sum T_r^i \times C_r^i = \sum T_r^i \times (C_k^i) / (B_k^i)$$

(2)

where $RI$ represents the potential ecological hazard index of various heavy metals in soil; $E_r{}^i$ is the potential ecological hazard coefficient of type $i$ heavy metal; $C_k{}^i$ is the concentration (mg/kg) of type $i$ heavy metal; $B_k{}^i$ is the parameter ratio of type $i$ heavy metal; and $T_r{}^i$ is the toxicity coefficient of type $i$ heavy metal. In this study, the toxicity coefficients of Hg, Cd, As, Ni, Cu, Pb, Cr and Zn are 40, 30, 10, 5, 5, 5, 2 and 1, respectively [30]. The pollution degree of heavy metals in the sediments corresponding to the value of $E_r{}^i$ and $RI$ is shown in Table 1 [29].

**Table 1.** The classification criteria of the potential ecological risk index.

| $E_r^i$ | $RI$ | Potential Ecological Risk |
|---|---|---|
| $E_r^i \leq 40$ | $RI \leq 150$ | Light ecological hazards |
| $40 < E_r^i \leq 80$ | $150 < RI \leq 300$ | Moderate ecological hazards |
| $80 < E_r^i \leq 160$ | $300 < RI \leq 600$ | Relatively strong ecological hazards |
| $160 < E_r^i \leq 320$ | | Strong ecological hazards |
| $E_r^i > 320$ | $RI > 600$ | Extremely strong ecological hazards |

## 3. Results and Discussion

### 3.1. Distribution Characteristics of Nitrogen and Phosphorus

Urban sewage discharge, surface runoff and hydrobiological debris in the lakes partly leads to the gradual accumulation of nutrients in the sediment of the lake, then forms the internal load of nutrients in the inland lake [31]. The total nitrogen (TN) content in the surface sediments of Yueliang Lake ranged from 1004 mg/kg to 3998 mg/kg, with an average of 2305 mg/kg. The total phosphorus (TP) content ranged from 264–728 mg/kg, with an average

of 530 mg/kg. Compared with other inland lakes in northern China [32–34], the *TN* and *TP* in the sediments of Yueliang Lake are at moderate and low levels, respectively.

Figure 2 shows the planar distribution characteristics of the *TN* and *TP* in surface sediments of Yueliang Lake. The planar distribution characteristics of *TP* are consistent with *TN*. The *TP* in the northwest is significantly higher than that in the southeast, and the high values are relatively concentrated in the central and western regions.

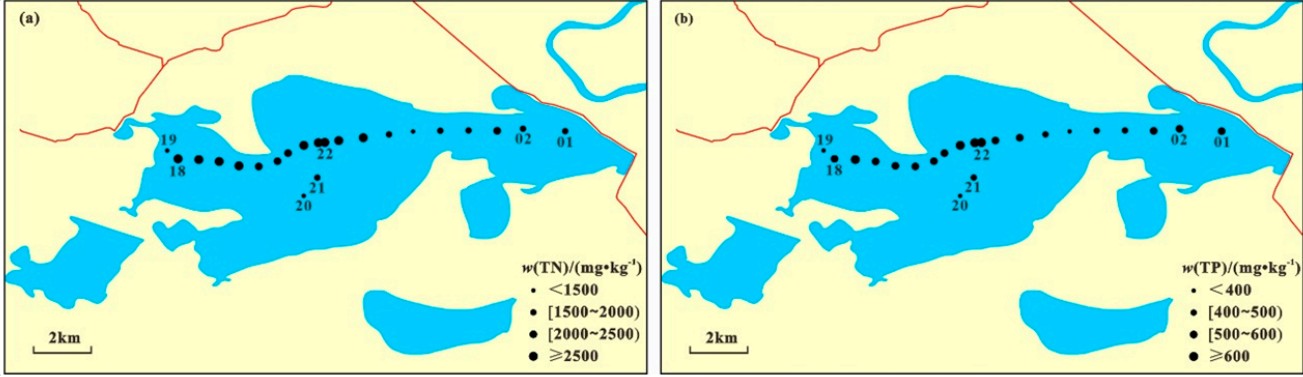

**Figure 2.** Distribution of *TN* (**a**) and *TP* (**b**) in surface sediments of Yueliang Lake.

### 3.2. Distribution Characteristics of Organic Carbon and Nutrient Ratio

Organic matter in sediments is the active substance for the adsorption, distribution and complexation of heavy metals, organic matter and other pollutants, and it is also an important indicator reflecting the organic nutrition degree of sediments [35]. The content of organic matter in the surface sediments of Yueliang Lake ranges from 0.26% to 4.04%, with an average value of 2.17%. On the planar distribution, organic matter also shows the characteristics of being high in the northwest and low in the southeast (Figure 3a). Most of the high value areas are concentrated in the middle west part of the lake, which is similar to the distribution of nitrogen and phosphorus. This further validates the above conclusion that the distribution characteristics of nitrogen and phosphorus are related to the content of organic matter.

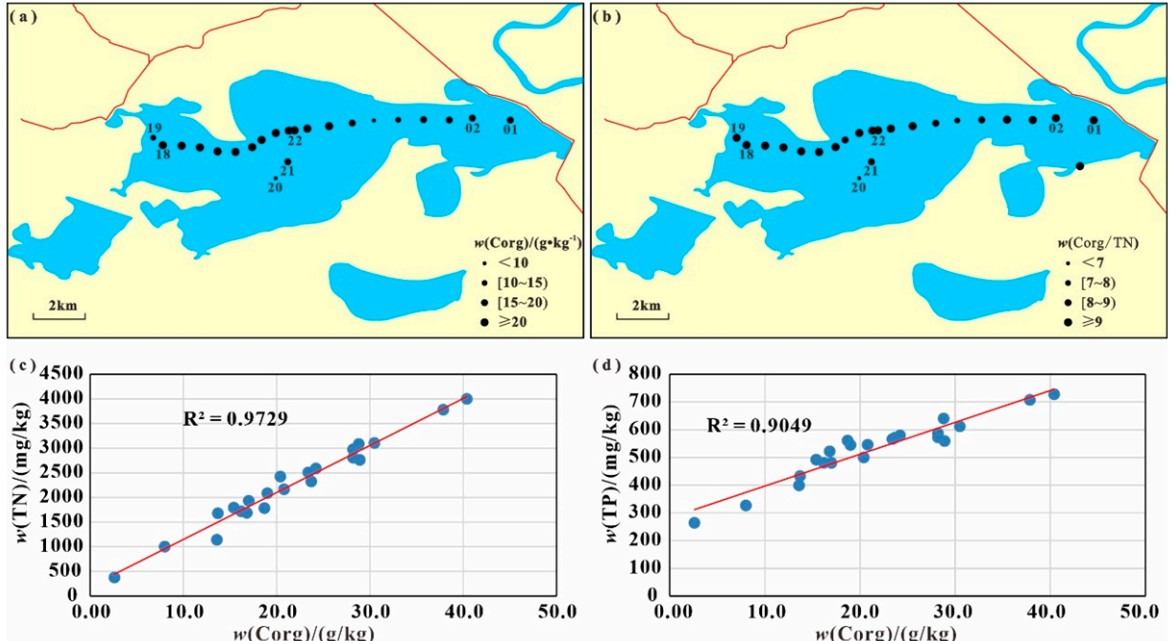

**Figure 3.** Distribution diagram of Corg (**a**), Corg/*TN* (**b**) and the scatter plot of Corg-*TN* (**c**) and Corg-*TN* (**d**) in surface sediments of Yueliang Lake.

In general, due to the water gate of Nenjiang, which is located on the east side of Yueliang Lake (Figure 1b), the water in the east has high fluidity, which is not conducive to the preservation and release of nitrogen and phosphorus in the bottom sediments. At the same time, due to the influence of surface runoff from farmland around the lake, many pollutants from cultivated farmland enter the lake, which in turn leads to high levels of organic matter, nitrogen and phosphorus in the sediments; some high content areas may also be affected by farmland planting methods [36].

To a certain extent, the ratio of organic matter to nitrogen (nutrient ratio) in lake sediments can reflect the difference in the source of organic matter for the sediments [35]. The lower the nutrient ratio means that the organic matter mainly comes from endogenous sources, such as algae and low aquatic plants that contain more protein, and in these cases the nutrient ratio is generally less than 7; the higher the nutrient salt ratio means that the organic matter is mainly from exogenous input, for example, the nutrient ratio of vascular terrestrial plants is usually greater than 20 [37,38]. The nutrient ratio of the surface sediments of Yueliang Lake ranged from 6.90 to 11.92, with an average value of 9.43. Figure 3b shows that the organic matter in the northwest part of Yueliang Lake is significantly higher than that in the central and eastern parts, indicating that the organic matter in the sediments of Yueliang Lake is a mixture of endogenous aquatic plants and exogenous terrestrial plants, and the northwest part has more endogenous while the central and eastern part has more exogenous.

There is a good positive correlation between nitrogen, phosphorus and organic matter in the surface sediments (Figure 3c,d), and the correlation coefficients were 0.9729 and 0.9049, respectively. The results indicate that there is homology between TN and organic matter in sediments, because nitrogen and carbon are organic components of organisms, and their contents are relatively constant in organisms. There is a high correlation between phosphorus and organic matter, which indicates that the phosphorus in surface sediments of Yueliang Lake mainly exists in organic form, and phosphorus may also come from anthropogenic activities.

### 3.3. Distribution Characteristics of Heavy Metals

#### 3.3.1. Heavy Metal Concentration of Surface Sediments

Figure 4 and Table 2 present the concentration bar charts of heavy metals (Hg, Cd, As, Ni, Cu, Pb, Cr and Zn) in the surface sediment collected from Yueliang Lake. The background values refer to the concentrations of heavy metals in the Songnen Plain soil and the soil eco-geochemical baseline of the alluvial plain of eastern China [39,40]. The concentrations of heavy metal elements ranged from 0.025 to 0.131 for Hg, 0.05 to 0.33 for Cd, 3.25 to 10.31 for As, 15.20 to 34.10 for Ni, 3.80 to 28.50 for Cu, 16.70 to 74.70 for Pb, 14.80 to 112.79 for Cr and 28.00 to 99.30 for Zn with all units of mg/kg, respectively. The average concentrations of the studied heavy metals descended in the order of Zn (71.08 mg/kg) > Cr (56.70 mg/kg) > Pb (53.69 mg/kg) > Ni (25.94 mg/kg) > Cu (18.67 mg/kg) > As (7.51 mg/kg) > Cd (0.15 mg/kg) > Hg (0.070 mg/kg). The comparison of heavy metal concentrations between Yueliang Lake and major lakes in the Songnen Plain and China showed that the concentration of heavy metals in the current study was lower than those in the major lakes (Table 1) [41,42], while higher than those in the Nen River, which is the largest tributary of the Songhua River [43]. The variation tendency of heavy metal concentrations was the same as that in Chagan Lake flowing into the Nen River, but different from that in the Songhua River [44,45]. The concentration of Pb in Yueliang Lake was much higher than that of the Nen River, Songhua River and Jingpo Lake, with a lower Cd, Ni and Zn concentration compared to Jingpo Lake and Songhua River [43,45,46]. The concentration of Cu was lower than that of the selected lakes, and slightly higher than the background value.

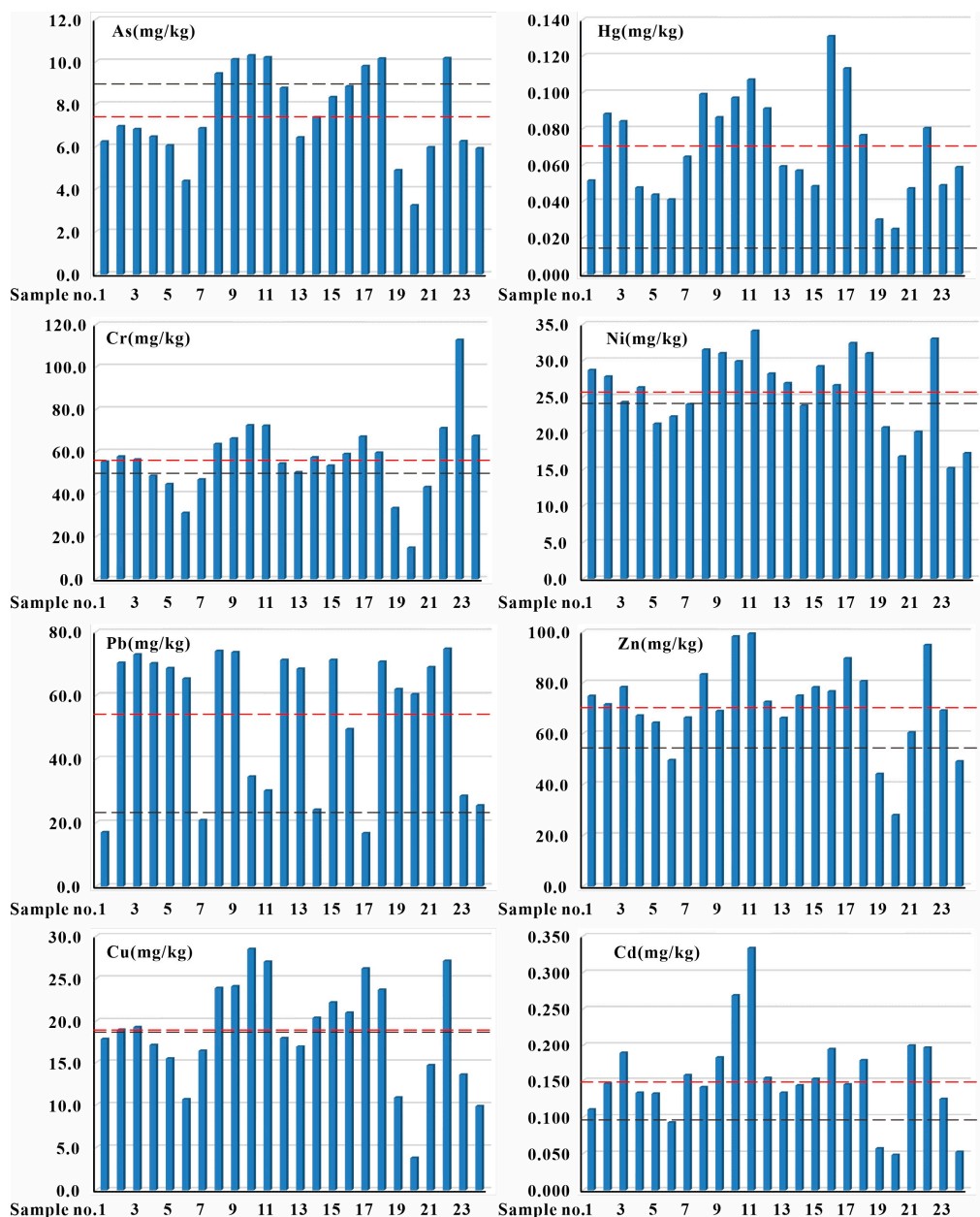

**Figure 4.** Histogram of heavy metal contents in the surface sediment of Yueliang Lake.

**Table 2.** Heavy metal concentrations in the surface sediments from Yueliang Lake and other selected lakes.

| Name of the Lake | Heavy Metal Concentrations (mg/kg) | | | | | | | | References |
|---|---|---|---|---|---|---|---|---|---|
| | Hg | Cd | As | Cr | Ni | Cu | Pb | Zn | |
| Yueliang Lake | 0.07 | 0.15 | 7.51 | 56.70 | 25.94 | 18.67 | 53.69 | 71.08 | This Study |
| Chagan Lake | 0.07 | 0.2 | 10.01 | 57.6 | 29.68 | 20.73 | 26.56 | 66.83 | [44] |
| Songhua River | 0.1 | 0.9 | 18.9 | 41.2 | 99 | 44.5 | 13.3 | 107 | [45] |
| Nen River | 0.027 | 0.24 | 5.2 | 26 | 24 | 21 | 5.4 | 54 | [43] |
| Jingpo Lake | 0.113 | 0.48 | 7.28 | 82.8 | 39.3 | 22.4 | 12.1 | 84.6 | [46] |
| Major Lakes in Songnen Plain | - | - | - | - | 35.07 | 29.09 | 25.57 | 189.78 | [42] |
| Major Lakes in China | 0.076 | 0.497 | 16.39 | 6.29 | 31.81 | 36.89 | 35.37 | 99.52 | [41] |
| Background Value | 0.015 | 0.099 | 9 | 50 | 24 | 18 | 22 | 54 | [39,40] |

### 3.3.2. Concentrations and Distribution of Heavy Metals in Sediment Cores

The upper part of the sediment core in Yueliang Lake is a dark gray sand-bearing clay layer, and the middle and lower part is a mainly gray to dark gray clay sand layer, with

rust spots and a small amount of sand spots and sand veins. Compared with the sediment core of Chagan Lake, the amount of sand spots is relatively less in the sediment core of Yueliang Lake [45]. Sediment core 1 was located in the center of the Yueliang Lake, where the sediment source and sedimentary environment are relatively stable, and the layering is obvious (Figure 5). Sediment core 2 was close to the Nenjiang sluice, in an area which receives a lot of exogenous deposits, so the grain size and composition of the sediments vary greatly, yellow gray with a gray black sandy clay layer and a blue gray to dark gray sandy clay layer formed in the middle (Figure 6).

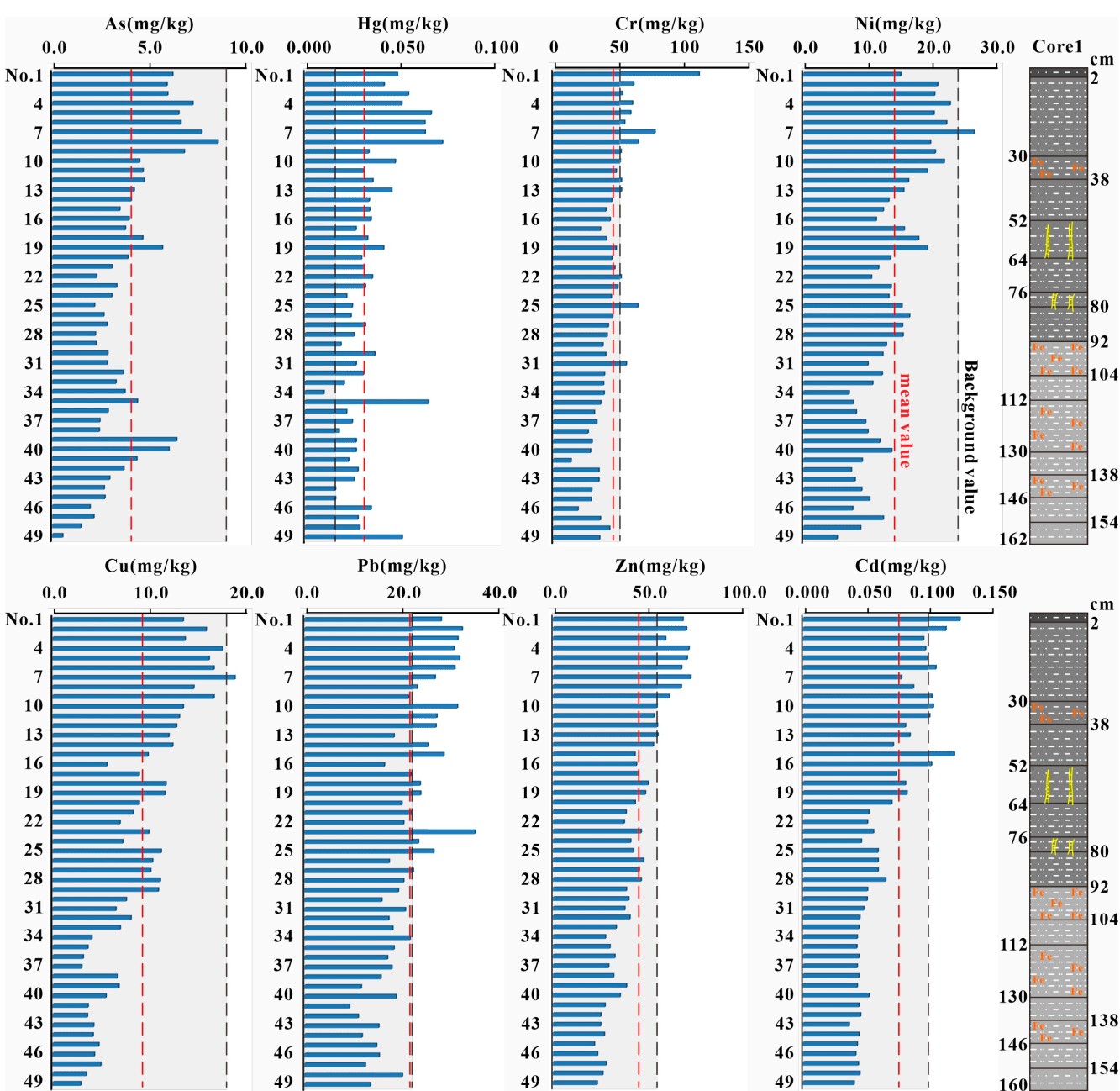

**Figure 5.** Content profiles of heavy metals in sediment core 1 from Yueliang Lake.

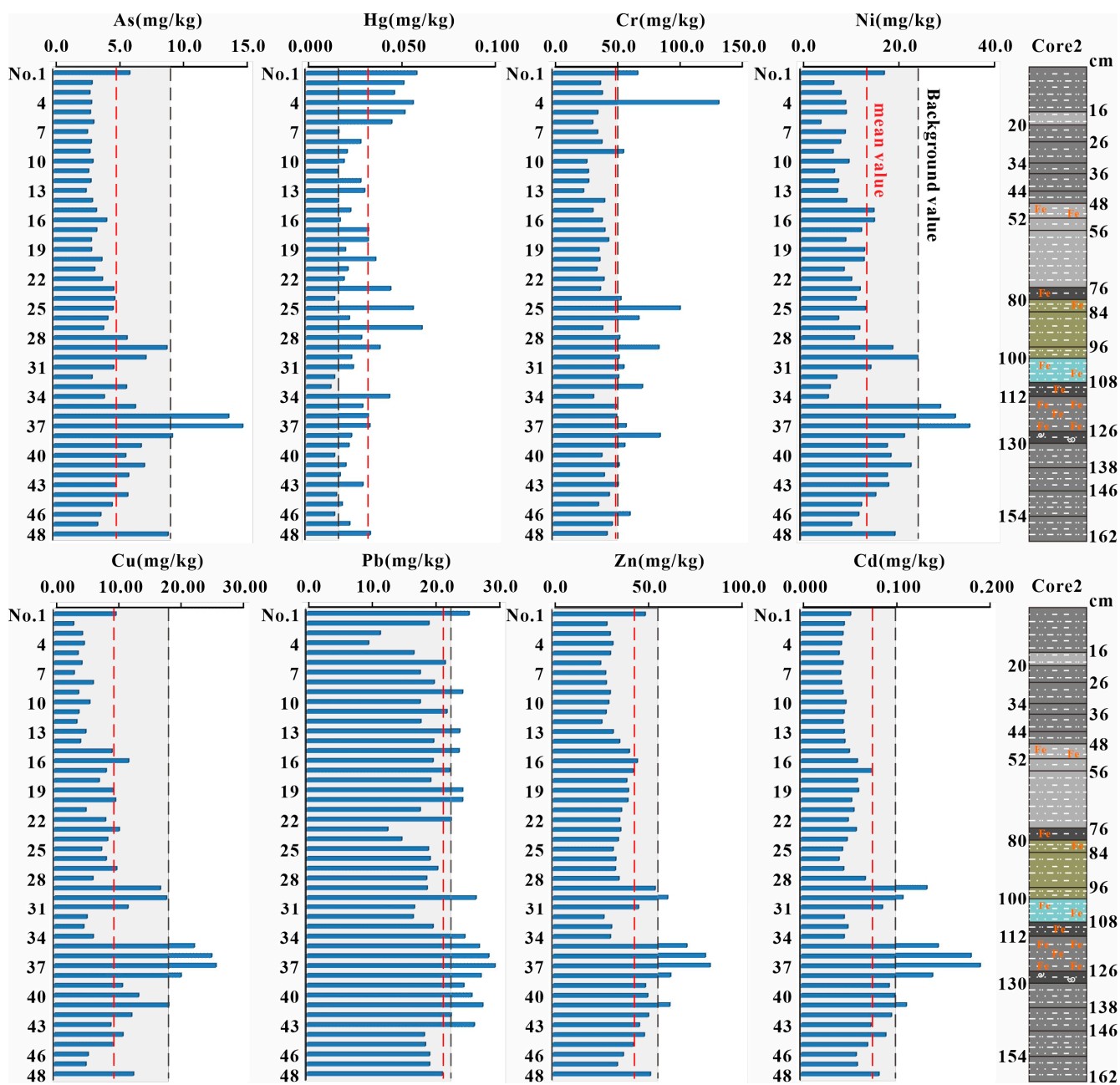

**Figure 6.** Content profiles of heavy metals in sediment core 2 from Yueliang Lake.

Table 3 shows the content characteristics of heavy metals in the vertical direction for Yueliang Lake. Except for Hg, the average content of the other elements is lower than the background value, and the coefficient of variation is less than 1, indicating that there is no significant accumulation of heavy metals in the vertical direction. From the vertical changes of element contents in the two sediment cores (Figures 5 and 6), the variation range of element contents in sediment core 2 is significantly greater than that in sediment core 1. In sediment core 1, the variation trend of the contents of heavy metals is similar, indicating that the sedimentary environment near the center of the lake is relatively stable. The concentrations of As, Ni, Cd, Cu, Pb and Zn in sediment core 2 increase significantly at 112–130 cm and 95–105 cm, which may be related to flood events in Yueliang Lake and its basin, such as the Second Songhua River flood in 1909 [25]. A flood carries more sediment, resulting in an increase in the input of elements. Moreover, related to the social environment at that time, the local government established the county system to pacify

banditry in the 1920s~1930s and animal husbandry and agriculture developed to a certain extent on the grassland. Blind grass destruction and deforestation reclamation aggravated the intensity of wind erosion and water erosion. The land salinization and soil erosion near the upper reaches of the lake are serious, and a large amount of sediment flows into the lake and is rapidly deposited in Yueliang Lake [25,44].

**Table 3.** Statistical table of the contents of heavy metals in the sediment cores of Yueliang Lake.

| Items | As | Hg | Cd | Cr | Ni | Cu | Pb | Zn |
|---|---|---|---|---|---|---|---|---|
| Minimum [#] | 0.50 | 0.01 | 0.04 | 13.60 | 5.26 | 2.91 | 9.30 | 21.90 |
| Maximum [#] | 8.60 | 0.07 | 0.13 | 113.00 | 26.70 | 19.00 | 35.50 | 73.20 |
| Mean [#] | 4.05 ± 1.80 | 0.03 ± 0.01 | 0.07 ± 0.03 | 45.15 ± 15.80 | 13.99 ± 4.97 | 9.21 ± 4.53 | 21.56 ± 6.47 | 44.08 ± 14.86 |
| Coefficient of variation [*] | 0.44 | 0.42 | 0.38 | 0.35 | 0.36 | 0.49 | 0.30 | 0.34 |
| Minimum [*] | 2.50 | 0.01 | 0.04 | 24.10 | 4.03 | 3.07 | 9.70 | 25.30 |
| Maximum [*] | 14.80 | 0.06 | 0.19 | 133.00 | 35.20 | 25.90 | 29.50 | 83.80 |
| Mean [*] | 4.84 ± 2.66 | 0.03 ± 0.01 | 0.07 ± 0.04 | 48.97 ± 20.18 | 13.44 ± 6.87 | 9.39 ± 5.74 | 20.99 ± 4.38 | 41.23 ± 13.87 |
| Coefficient of variation [*] | 0.55 | 0.45 | 0.52 | 0.41 | 0.51 | 0.61 | 0.21 | 0.34 |
| Background value | 9 | 0.015 | 0.099 | 50 | 24 | 18 | 22 | 54 |

Notes: [#] stands for sediment core 1; [*] stands for sediment core 2.

*3.4. Ecological Risk Assessment*

3.4.1. Evaluation of Potential Ecological Risk Index

Table 4 shows the potential risk index results of eight heavy metals in the surface sediments of Yueliang Lake. The order of the average value of the single potential risk index is Hg (186.60) > Cd (46.34) > Pb (12.20) > As (8.06) > Ni (5.40) > Cu (5.19) > Cr (2.27) > Zn (1.32), which indicates that Hg poses strong ecological hazards, Cd poses moderate ecological hazards, and the other six heavy metals are light ecological hazards in the surface sediments. Hg and Cd are the most important hazard factors in the study area: Cd pollution is not high, and 75% of the samples show light ecological hazards from it, but there is a certain ecological hazard due to its strong toxicity. In addition, 46.8% of the samples are $E_i > 160$ for Hg, indicating strong ecological hazards. The *RI* of surface sediments is 104.34~444.21, with an average value of 267.37, which reflects that the heavy metals in the sediments of Yueliang Lake are at the level of moderate ecological hazards in general, and about 54.2% of the samples are moderate ecological hazards.

For the sediment core, the $E_i$ average value of Cd, Pb, As, Ni, Cu, Cr and Zn is less than 40, which belongs to the level of light ecological hazards. The $E_i$ of Hg ranges from 26.67 to 194.99, with an average value of 86.17. Only 6% of the samples have an $E_i$ greater than 160, and the samples where $E_i$ is between 80 and 160 account for about 43.5%, indicating moderate ecological hazards. The *RI* of the sediment core is 56~252.70, with an average value of 127.66, which reflects heavy metals of sediments in the vertical direction being at the light ecological hazard level.

3.4.2. Evaluation of Potential Ecological Risk Index

Substituting the measured heavy metals data into Equation (1) enables the $I_{geo}$ to be obtained for the surface sediments of Yueliang Lake. The $I_{geo}$ ranges of As, Hg, Cr, Ni, Cu, Pb, Zn and Cd elements are −0.25~−0.39, 0.15~−2.54, −2.34~0.59, −1.24~−0.08, −2.83~0.08, −0.98~1.18, −1.53~0.29 and −1.62~1.17, respectively (Figure 7). The mean arithmetic values of heavy metals were Hg > Pb > Cd > Zn > Cr > Ni > Cu > As, and the average values of the $I_{geo}$ of As, Cu, Cr, Ni, Zn and Cd are less than 0. Only a few samples are even slightly above 0, indicating that the surface sediments are not polluted by these six heavy metals. The average $I_{geo}$ of Pb is 0.54, between 0 and 1, indicating slight pollution. The average $I_{geo}$ of Hg is 1.52, which belongs to moderate pollution; it is worth noting that 58% of the samples show moderate pollution (between 1 and 2), and 25% of the samples have $I_{geo}$ greater than 2, showing heavy pollution, mainly distributed in the central and western parts of Yueliang Lake.

**Table 4.** Calculation results of the $E_i$ and $RI$ in the surface sediments and sediment core of Yueliang Lake.

| Items | $E_i$ | | | | | | | | $RI$ |
|---|---|---|---|---|---|---|---|---|---|
| | As | Hg | Cd | Cr | Ni | Cu | Pb | Zn | |
| Minimum [&] | 0.06 | 66.67 | 0.59 | 3.17 | 1.06 | 3.80 | 0.52 | 14.67 | 104.34 |
| Maximum [&] | 11.46 | 349.33 | 4.51 | 7.10 | 7.92 | 16.98 | 1.84 | 101.11 | 444.21 |
| Mean [&] | 8.06 ± 2.84 | 186.60 ± 74.15 | 2.27 ± 0.73 | 5.40 ± 1.12 | 5.19 ± 1.72 | 12.20 ± 4.98 | 1.32 ± 0.31 | 46.34 ± 19.20 | 267.37 ± 92.35 |
| Minimum [*] | 0.59 | 26.67 | 1.36 | 0.84 | 0.81 | 2.11 | 0.41 | 11.11 | 56.00 |
| Maximum [*] | 16.42 | 194.99 | 13.26 | 7.32 | 7.19 | 8.08 | 1.55 | 57.99 | 252.70 |
| Mean [*] | 4.93 ± 2.53 | 86.17 ± 37.83 | 4.70 ± 1.81 | 2.86 ± 1.24 | 2.58 ± 1.43 | 4.83 ± 1.25 | 0.79 ± 0.27 | 20.78 ± 9.53 | 127.66 ± 44.31 |

Notes: [&] stands for surface sediment; [*] stands for sediment core.

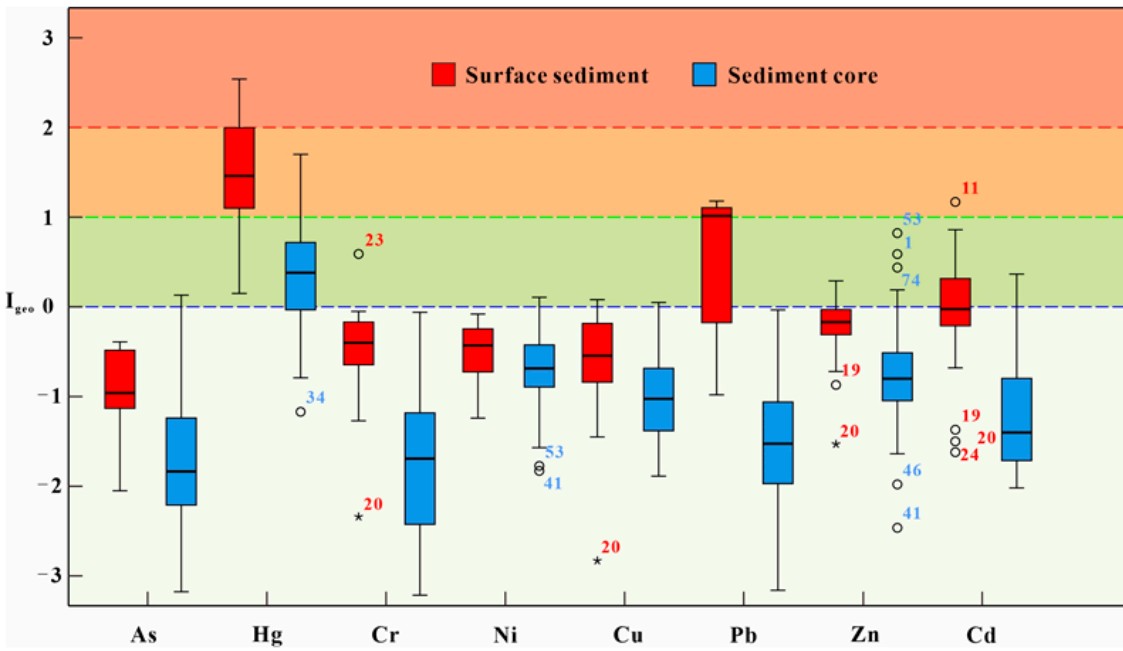

**Figure 7.** Box plot of Igeo in the surface sediments and sediment core of Yueliang Lake.

The average $I_{geo}$ values of heavy metals in the sediment core show Hg > Ni > Zn > Cu > Cd > Pb > As = Cr. Other than Hg ($I_{geo}$ = 0.39), all other elements are less than 0, indicating that the sediment is basically free from pollution except Hg in the vertical direction. The $I_{geo}$ of Hg ranges from −1.17 to 1.70, and 73.2% of the samples have $I_{geo}$ greater than 0, indicating slight pollution.

### 3.5. Source Identification

Figure 8 shows that the correlation between heavy metals is quite different in different sediments of Yueliang Lake, reflecting the inconsistency between the sources of heavy metals in surface sediments and sediment column sediments. Figure 8a shows that Hg and Cd are basically unrelated to other elements in the surface sediments; the correlation coefficients between Pb, Cu and Zn are greater than 0.6, which belongs to strong correlation. The correlation coefficients of Ni and Cr are 0.43, which is related. The correlation coefficients between As, Cr and Pb are greater than 0.5, which is strong correlation. These results indicate that these elements have similar geochemical properties and the same source or generate compound pollution in the surface sediments of Yueliang Lake. The correlation between heavy metal elements in the sediment core is not strong (Figure 8b). Only the correlation coefficients of Cu and Hg (0.57), Cu and Ni (0.50), Cr and Ni (0.46) and Hg and Ni (0.44) are greater than 0.4, which shows a certain correlation, indirectly reflecting the influence of soil forming parent materials for the enrichment of different heavy metals.

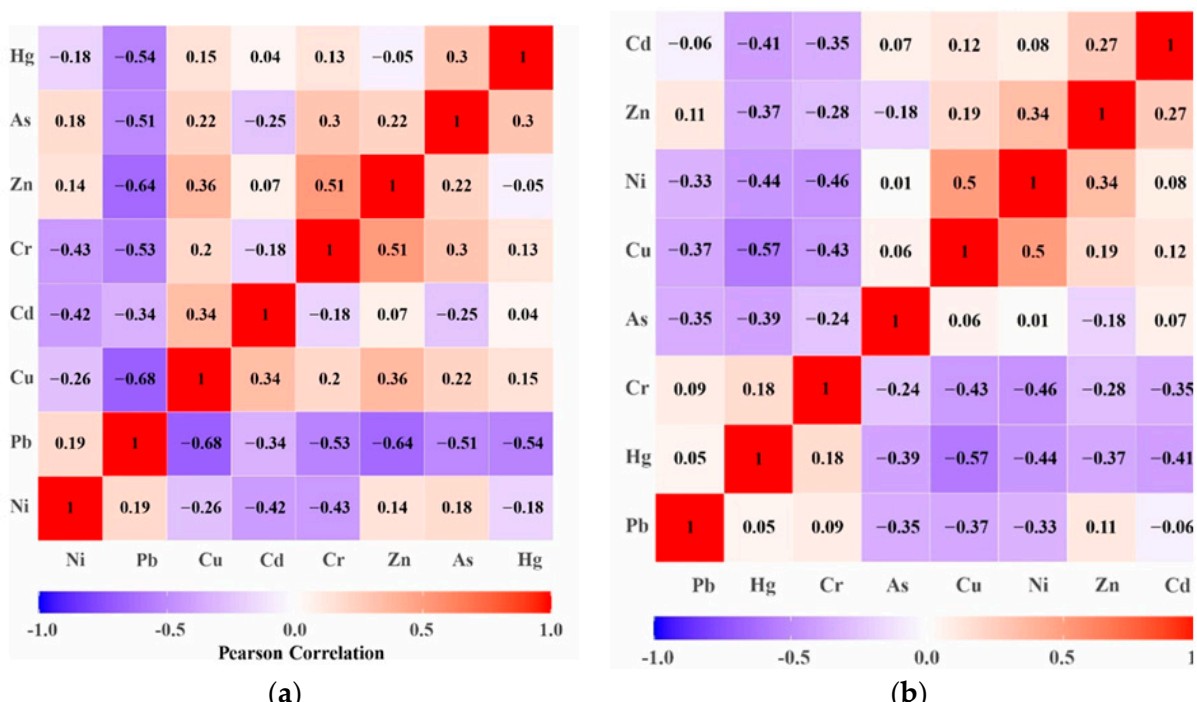

**Figure 8.** Correlation coefficient heat map of heavy metals in the sediments of Yueliang Lake (**a**) the surface sediments, (**b**) the sediment core.

The Bartlett sphericity test (0.00 < 0.05) and KMO measurement test (0.766 > 0.6) were carried out for the dataset of heavy metals in the surface sediments of Yueliang Lake. The results show that the PCA is suitable for the dataset. The results of the PCA show that the eight elements are divided into three categories (Figure 9), and the cumulative variance contribution rate reaches 77.53%.

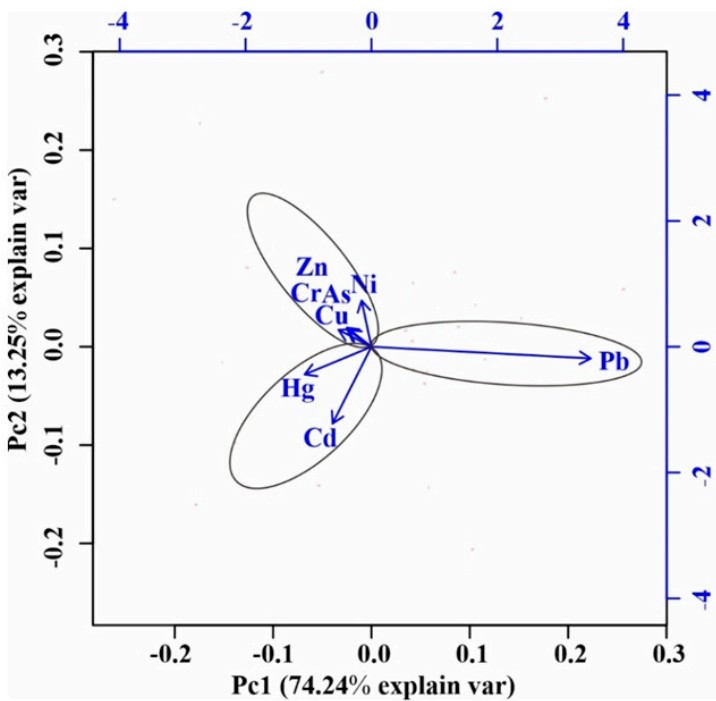

**Figure 9.** PC1 and PC2 biplot of the PCA in the surface sediments from Yueliang Lake.

The variance contribution rate of the first principal component (F1) is 45.78%. The heavy metals with high load are Cr, Ni, Cu, Zn and As, which represent natural sources and are related to the weathering process of parent rock, the soil forming process and other natural processes. Figure 10 shows the changes of Cr and Ni with $Al_2O_3$ and $Fe_2O_3$ in the surface sediments. The geochemical behavior of Cr, Ni, Zn and Cu elements is mainly controlled by the content of clay minerals. The main sources of heavy metals are the rocks and soils around Yueliang Lake. They are deposited or adsorbed on surface sediments in the form of clay minerals and sulfides. The migration ability of As is relatively weak, and the potential ecological risk of the sediments of Yueliang Lake is low or zero. Considering the low temperature in Northeast China, scattered farmers around the lake area burn coal for heating under extreme weather conditions, which can also be regarded as the main source of As in the surface sediments.

The variance contribution rate of the second principal component (F2) is 16.29%, and the heavy metals with higher loading are Hg and Cd. Its concentrations are higher than the background values and are considered major contributors to potential ecological risks. The evaluation of the geo-accumulation index indicates that the pollution level of Cd and Hg is moderate to slight in the surface sediments of Yueliang Lake. The main sources of Hg include the discharge of Hg-containing wastewater, the atmospheric deposition of Hg-containing particles in the atmosphere and surface runoff. Most of the Hg is retained in the sediments [47]. In addition, Yueliang Lake is the largest lake in Daan County and an important fishery base in Jilin Province. Previously, the addition of a large amount of exogenous feed and pesticides led to the final deposition of heavy metals in the sediments of the lake bottom. Studies have shown that fish meal in aquatic feeds contains a certain amount of Cd [48]. Cd and Hg in the surface sediments of Yueliang Lake may come from the bait added during aquaculture, fish excreta, atmospheric deposition of Hg-containing particles and surface runoff, pesticide residues, etc.

The variance contribution rate of the third principal component (F3) is 15.464%, and the only heavy metal with high load is Pb. Previous studies have shown that transportation (vehicle exhaust, tire wear, brake wear) is the key source of Pb pollution in sediments [49,50]. In addition, coal mining and combustion emissions may be important contributors of Pb in atmospheric deposition [51]. Therefore, the Pb pollution may be related to transportation

and coal-related industrial activities in the study area. The average Igeo of Pb is 0.54 in the surface sediments of Yueliang Lake, indicating light pollution. With the development of local tourism and the rapid increase in traffic flow, Pb pollution should be paid sufficient attention.

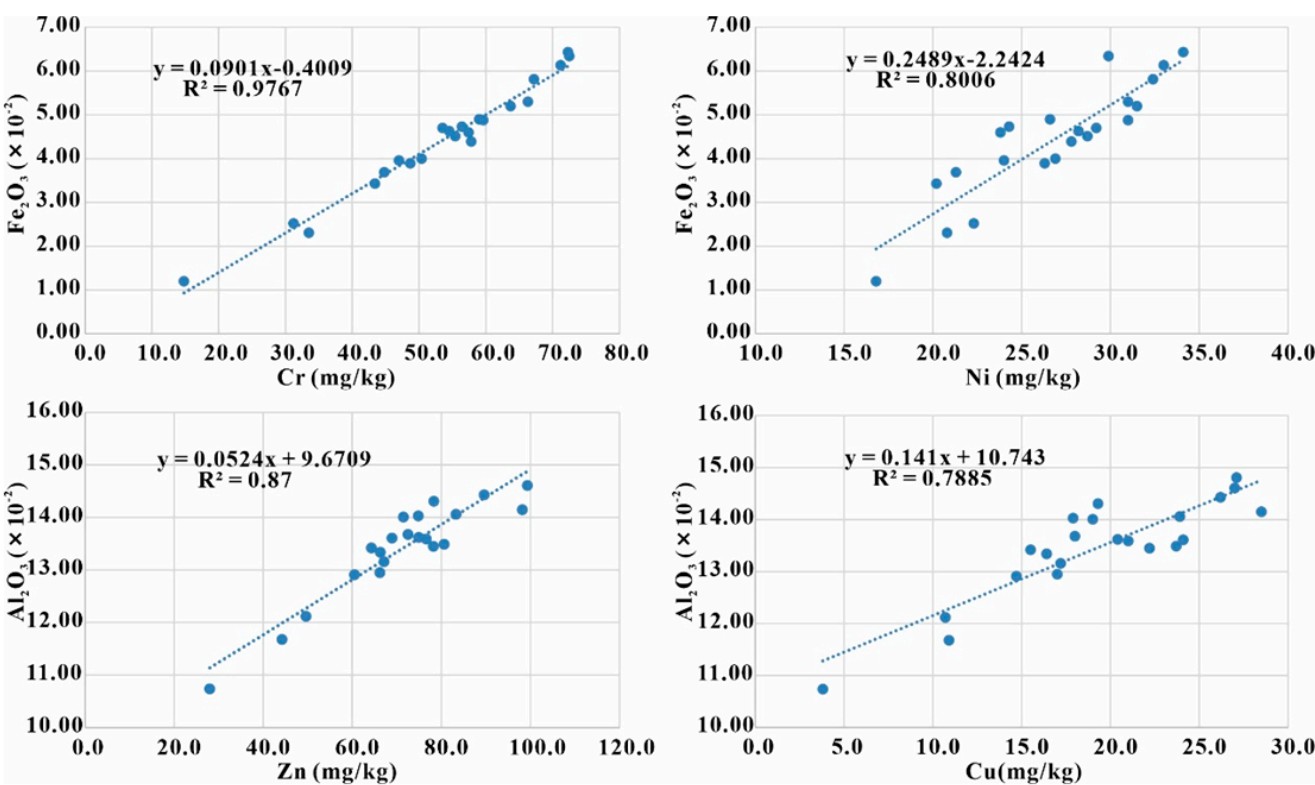

**Figure 10.** Distribution diagram of Cr, Ni, Zn and Cu (mg/kg) versus $Al_2O_3$ and $Fe_2O_3$ (wt%) in the surface sediments of Yueliang Lake.

## 4. Conclusions

Compared with other inland lakes in northern China, the contents of nitrogen and phosphorus in the sediments of Yueliang Lake are at a low level. The nutrient salt ratio is 6.90~11.92, with an average of 9.43, which shows that the organic matter is a mixture of endogenous aquatic plants and exogenous terrestrial plants in the sediments of Yueliang Lake. Except for Cd and Hg, most elements are not at a polluted level in the surface sediments. The *RI* value is 104.34~444.21, and the average value is 267.37. The ecological risk of heavy metals is at the level ofmoderate-slight ecological hazards. In the vertical direction, the sediments are at the level of slight ecological hazards, and the other heavy metals are basically not at a polluted level except Hg. The results of the CA and PCA show that in the sediments of Yueliang Lake they come mainly from three sources: Cr, Ni, As, Zn and Cu are mainly from natural sources; the main sources of Cd and Hg are highly related to agricultural activities and energy development; and Pb is coming from vehicle traffic and coal-related industrial activities. Therefore, the prevention of Hg and Cd pollution should be emphasized in the future development of the Yueliang Lake area.

**Author Contributions:** Writing—original draft preparation, J.Z. and P.L.; software and data processing, C.Z. and P.L.; writing—review and editing, M.W., Y.L. and L.H.; collecting the samples, Y.L. and S.C.; funding acquisition, P.L., L.H. and M.W. All authors have read and agreed to the published version of the manuscript.

**Funding:** This work was supported by the National Natural Science Foundation of Guangxi (2022GXNSFBA035548 and 2021JJA150037) and the National Natural Science Foundation of China (42203067); and the Doctoral Research Foundation of Guilin University of Technology (GUTQDJJ-2019166).

**Acknowledgments:** The authors want to express their gratitude to the Institute of Environment and Sustainable Development in Agriculture, Chinese Academy of Agricultural Sciences (CAAS).

**Conflicts of Interest:** The authors declare no conflict of interest.

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
