# Peer review of "Distribution Characteristics and Ecological Risk Assessment of Nitrogen, Phosphorus, and Some Heavy Metals in the Sediments of Yueliang Lake in Western Jilin Province, Northeast China"

_water, doi:10.3390/w14203306_

Round 1
Reviewer 1 Report
The current manuscript entitled “Distribution Characteristics and Ecological Risk Assessment of Nitrogen, Phosphorus, and Heavy Metals in the Sediments of Yueliang Lake in Western Jilin Province, Northeast China” by Zhang et al. is well written and described. The study was carried out carefully with a sufficient sample collection and analyzed using appropriate tools such as Igeo and risk indices. Overall contents and quality of the manuscript are fine and I only suggest minor improvements. My specific comments are:
1. The abstract is too long. Try to reduce it to <250 words and remove the unnecessary discussion.
2. All models/equations should be written using the MS Word equation tool. They are provided as images.
3. Igeo: geo should be subscripted.
4. Improve the table/figure legends.
5. Fig. 1: The map of China (a) looks of low quality as compared to b and c. Try to improve it.
6. The conclusion section should be rewritten. Only, major outcomes, shortcomings, recommendations, and future scope of the work should be provided.
7. References: should be updated to recent.
Author Response
We would like to thank all the reviewers for helpful comments. We have addressed all the comments and questions by the reviewer. The red font is the modified part.

Reviewer 2 Report
OVERALL: - Please, enlarge and re-arrange font sizes to guide the reader properly in all sections. All figrues must be composed of HD images. It is mandatory to improve the scientific quality of the whole manuscript.
- Please, pay attention to the JOURNAL TEMPLATE in all sections, including tables, references, captions, units, equations, and Figures.
- Please, insert all corrected axes and labels, in order to guide the reader properly in the understanding of the whole manuscript. In addition, please improve the contrast between colours.
INTRODUCTION: Please, introduce in the scientific background of your study the importance of wetlands in the management of vegetated water resources (i.e.,
Lama, G.F.C., Errico, A., Pasquino, V., Mirzaei, S., Preti, F., Chirico, G.B. 2022. Velocity uncertainty quantification based on Riparian vegetation indices in open channels colonized by Phragmites australis. J. Ecohydraulics 7(1), 71–76. https://doi.org/10.1080/24705357.2021.1938255
Müllerová, J., Gago, X., Bučas, M., Company, J., Estrany, J., Fortesa, J., Manfreda, S., Michez, A., Mokroš,M., Paulus, G., Tiškus, E., Tsiafouli, M.A., Kent, R. (2021). Characterizing vegetation complexity with unmanned aerial systems (UAS) – A framework and synthesis. Ecological Indicators, 131, 108156. https://doi.org/10.1016/j.ecolind.2021.108156).
METHODS: Please, insert a Figure for each sub-section. This will improve the scientific quality of your study, as a great support to all the equations proposed here.
DISCUSSION and CONCLUSIONS: These two sections must be re-arranged according to the suggestions indicated by the reviewer.
Author Response

(The authors gave the same response as above.)

Reviewer 3 Report
Title: “Distribution Characteristics and Ecological Risk Assessment of Nitrogen, Phosphorus, and Heavy Metals in the Sediments of Yueliang Lake in Western Jilin Province, Northeast China”
Several points are important to be addressed before going to accept this article.
1. The title should be improved. Add (and Some Heavy Metals in the …). Some relevant Keywords also need to be added
2. The introduction needs to improve with some recent publications. The author may use the following ref. https://doi.org/10.3390/jmse9121328 ; https://doi.org/ 10.3390/su13137077 ; https://doi.org/10.3390/ma15113922
3. In many sentences, English grammar and style need to improve.
4. Some Figs need to improve. Fig. 3C and D, Fig. 4 should contain SD or SE. Remove the Chinese words from Fig. 7. Improve the resolution of Eq. 1 and 2 or make it writable. In Tables 3 and 4, make the data as Mean ± SD or as Mean ± SE.
5. In Line 18: “and some heavy metals (As, Hg, Cd, Cr, Cu, Pb, Zn and Ni)”
6. IN Line 19: Remove to Line 17 move “(As, Hg, Cd, Cr, Cu, Pb, Zn and Ni)” to Line 17
7. Line 22: “analysis (PCA) methods”
These major comments need to resolve before accepting the manuscript.
Author Response

(The authors gave the same response as above.)

Round 2
Reviewer 2 Report
The article had been improved according to the indications of the reviewer. Just few corrections are needed:
OVERALL: Please, improve the contrast in Fig. 6 and Fig. 7. This is useful for a better clarity of the scientific message of your results.
INTRODUCTION: Please, consider in the scientific background of your study the importance of advanced statistical methods in geo-statistics for the management of water resources broadly speaking (i.e.,
Brunetti, G.F.A., Fallico, C., De Bartolo, S., Severino, G. (2022). Well‐Type Steady Flow in Strongly Heterogeneous Porous Media: An Experimental Study. Water Resour. Res., 58(5), e2021WR030717. doi.org/10.1029/2021WR030717.
Khan, M.A., Sharma, N., Lama, G.F.C., Hasan, M., Garg, R., Busico, G., Alharbi, R.S. 2022. Three-Dimensional Hole Size (3DHS) Approach for Water Flow Turbulence Analysis over Emerging Sand Bars: Flume-Scale Experiments. Water 14, 1889. https://doi.org/10.3390/w14121889).
Author Response
Thanks again to the reviewers and editors for their careful and patient review. We have addressed all the comments and questions by the reviewer.
